# How well does GPT-4o understand vision? Solving Standard Computer Vision Tasks with Multimodal Foundation Models

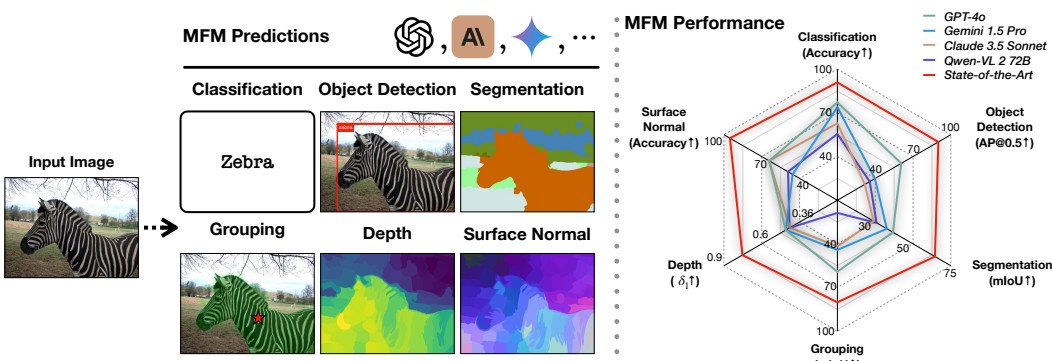

Figure 1: We solve standard semantic and geometric computer vision tasks using popular multimodal foundation models (MFMs) and established datasets. The left part of the figure displays GPT-4o's predictions for different tasks, including classification, object detection, semantic segmentation, grouping, depth prediction, and surface normal prediction. The right part of the figure quantifies the performance of MFMs on these tasks and provides comparisons with specialist state-of-the-art vision models for each task.

## Abstract

Multimodal foundation models, such as GPT-4o, have made remarkable progress recently. However, it is not clear exactly where these models stand in terms of understanding vision. In this paper, we **evaluate the performance of popular multimodal foundation models** (GPT-4o, Gemini Pro, Claude 3.5 Sonnet, Qwen2-VL) **at standard computer vision tasks** (semantic segmentation, object detection, image classification, depth and surface normal prediction) and **using established datasets** (e.g., COCO, ImageNet and its variants, etc).

The main challenges to performing this are: **1)** the models are trained to output text and cannot natively express versatile domains, such as segments or 3D geometry, and **2)** many of the leading models are proprietary and accessible only at an API level, i.e., there is no weight access to adapt them. We address these challenges by translating standard vision tasks into equivalent text-promptable and API-compatible tasks via prompt chaining.

We observe that **1)** the models are not close to the state-of-the-art at any tasks, and **2)** they perform semantic tasks notably better than geometric ones. However, **3)** they are respectable generalists; this is remarkable as they are presumably trained on only image-text-based tasks primarily. **4)** While the prompting techniques affect the performance, better models exhibit less sensitivity to prompt variations. **5)** GPT-4o performs the best, getting the top position in 5 out of 6 tasks.

## 1    INTRODUCTION

Multimodal foundation models (MFMs), such as GPT-4o, Gemini 1.5 Pro, and Claude 3.5 Sonnet (OpenAI, 2024; Reid et al., 2024; Anthropic, 2024), have gone far in recent months, with their demos appearing highly impressive (OpenAI, 2024). However, while the community has extensively investigated their remarkable language proficiency Hendrycks et al. (2020); Chen et al. (2021); Rein et al. (2023); Chiang et al. (2024), the extent of their vision capabilities is vague in comparison. We still lack a well-calibrated understanding of their performance on established vision tasks and datasets, particularly across diverse axes of vision, e.g. semantics, 3D, etc.

Most of the existing vision benchmarks of MFMs primarily target text (e.g., VQA) or tasks closely tied to text, like classification. (Yue et al., 2024; Fu et al., 2024; Tong et al., 2024b;a; Rahmanzadehgervi et al., 2024; Wu & Xie, 2024). While they provide useful insights, several key limitations persist. First, it is unclear how much solving these benchmarks truly depends on the visual input, and some were shown to mainly measure the language capabilities of MFMs while overlooking the vision component (Tong et al., 2024a). Second, they all require the model to output text, making it hard to compare the vision capabilities of MFMs against vision-only tasks and specialist models developed by the community. Third, they do not shed light on other aspects of visual understanding, such as 3D geometry, grouping, or segmentation, that are less text oriented.

We address these limitations by evaluating MFMs on well-established vision tasks and datasets developed by the community. Specifically, we test GPT-4o, Claude 3.5 Sonnet, Gemini 1.5 Pro, and Qwen2-VL on classification, object detection, semantic segmentation, grouping, depth prediction, and surface normal prediction using COCO, Hypersim, ImageNet and its variants (Lin et al., 2014; Roberts et al., 2021; Russakovsky et al., 2014). Most of these tasks, however, require dense pixel-wise predictions not readily compatible with the default text output of MFMs. To address this challenge, we split each task into multiple sub-tasks, each of which can be solved in a textual form via prompting (see Sec. 3). This results in a *prompt-chaining framework that can be applied to any MFMs with a text interface (e.g., ChatBot APIs) to solve standard vision tasks*. Specifically, our proposed approach allows MFMs to **1)** detect bounding boxes, **2)** generate complete segmentation masks for complex scenes, **3)** extract semantic entities from images similar to SAM (Kirillov et al., 2023b), **4)** estimate dense depth and surface normal maps. Please see Fig. 1 for an overview. This enables direct comparison with vision-only models, offering a holistic understanding of the vision capabilities of MFMs.

We find that MFMs achieve good performance in most cases and show respectable generalist abilities, with GPT-4o scoring the best in 5 out of 6 tasks. However, *they still lag behind task-specific state-of-the-art vision models in all tasks*. In particular, we find that the MFMs perform geometric tasks significantly worse than semantic ones. Furthermore, we perform a detailed prompt sensitivity analysis for each task and find the performance varies for different prompts, though better models exhibit less sensitivity. We will open-source our documented code to enable researchers to explore performant prompt chaining strategies for MFMs.

## 2    RELATED WORK

**Advances in MFMs.** There has been remarkable progress in MFMs (Alayrac et al., 2022; Wang et al., 2022; Team et al., 2023; Achiam et al., 2023; Li et al., 2023a; Dai et al., 2023; Bai et al., 2023; Liu et al., 2024; Beyer et al., 2024; Team, 2024; Wang et al., 2024; OpenAI, 2024; Anthropic, 2024; Reid et al., 2024) (see (Zhang et al., 2024; Yin et al., 2023) for surveys), leading to strong performance across a wide range of tasks that require joint vision and linguistic capabilities such as captioning, visual question answering, and instruction following. Despite the progress, it is unclear how well these models perform tasks that require dense visual understanding, which is our main focus.

**Benchmarking vision capabilities of MFMs.** Many works investigate the vision capabilities of MFMs by developing VQA-style benchmarks that combine visual and textual inputs to generate textual outputs (Liu et al., 2024; Li et al., 2023b; Fu et al., 2024; Tong et al., 2024b; Rahmanzadehgervi et al., 2024; Al-Tahan et al., 2024; Yue et al., 2024; Jiang et al., 2024; Tong et al., 2024a). While these approaches offer valuable insights, they are incompatible with traditional computer vision models, making direct comparisons difficult. In contrast, *we directly evaluate MFMs on standard vision tasks*, enabling direct comparison with strong vision specialists to track MFMs'

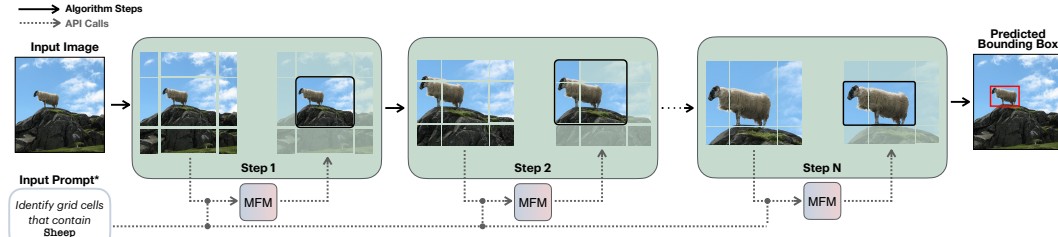

Figure 2: **Object detection algorithm.** At each step, we divide the image into a grid of crops, and each crop is queried for the presence of the target object (Sheep in the figure) through the model. Grid cells without the object are discarded, and the process is repeated until the full object is located. *This is a summary of the actual prompt. See full prompt in Appendix B.

progress. (Tong et al., 2024a) evaluates MFMs on vision datasets (Lin et al., 2014; Zhou et al., 2017; Brazil et al., 2023) by repurposing dataset annotations into text format. We differ by translating MFM outputs into the annotation format instead, e.g. segmentation maps. Crucially, this enables apples-to-apples comparisons with vision specialist models, using standard task-specific metrics, and qualitative analyses in the tasks' native output space.

**Prompting techniques for MFMs.** Various prompting techniques have been developed for MFMs (Wei et al., 2022; Zhou et al., 2022; Khot et al., 2022; Yao et al., 2024). We follow a similar strategy and decompose complex vision tasks into simpler sub-tasks that MFMs can handle. Several works developed prompting techniques to unlock vision capabilities of MFMs (Yang et al., 2023a; Wu et al., 2024; Hu et al., 2024; Wu & Xie, 2024). A related work is DetToolChain (Wu et al., 2024), which develops a prompting mechanism for object detection. We differ by **1)** focusing on a wider range of tasks including semantic and geometric ones **2)** for several MFMs including closed- and open-weight ones **3)** with a simpler yet effective and cost-efficient prompt chaining mechanism.

## 3 PROMPT CHAINING FOR SOLVING VISION TASKS WITH MFMS

In this section, we describe the developed prompt chaining techniques that enable MFMs to solve standard computer vision tasks, namely image classification, object detection, semantic segmentation, grouping, depth, and surface normal prediction. These techniques are based on the main idea of breaking the original task into multiple simpler sub-tasks that can be solved in a language format, e.g., identifying whether an object is present in a patch of an image. We then solve each sub-task by prompting an MFM. To guide the choice of how to split each task into sub-tasks, we rely on our early key observation that most MFMs are relatively strong at image classification (see, e.g., Tab. 1) and, therefore, try to split each task into multiple classification sub-tasks. We provide the pseudo-code for each technique in the Appendix.

**Image classification.** This task involves directly identifying the main class of an image from a set of classes. Here, the model is presented with a list of all ground-truth classes and tasked with assigning the image to the correct category. Following (Jiang et al., 2024), we group images into batches for efficiency, as we observed no significant decrease in accuracy when using this approach.

**Object detection.** In this task, the goal is to predict bounding box coordinates that tightly localize the objects in the image. Similar to Yang et al. (2023b), our initial attempts showed that many MFMs fail at predicting the coordinates directly. We, therefore, develop a prompt chaining method and divide the original task into two stages. The first stage has a single sub-task to identify all present objects in the image. In the second stage, for each object, we regress its coordinates via recursive zooming. Specifically, we divide the image into grid cells and ask the model to identify whether (a part of) the object is present in each cell. We then discard cells without objects, reducing the search space. We apply this process recursively, progressively eliminating irrelevant regions of the image until only the object of interest remains present in the image. We use two grid resolutions: a coarse grid for quick downsampling and a finer grid for precise edge refinement that allows us to reduce the number of steps. Please see Fig. 2 for an overview and Algorithm 2 for the pseudo-code.

**Semantic segmentation.** In this task, the goal is to assign one of the semantic classes to each pixel in an image. Instead of per-pixel querying, we split the image into pixel groups using an

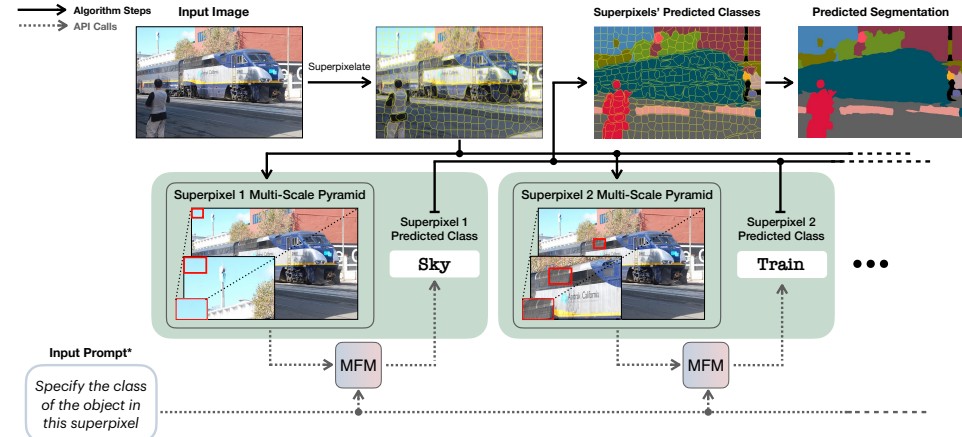

Figure 3: **Semantic segmentation algorithm.** We divide the image into superpixels and create "multi-scale pyramids" of superpixels. The pyramids are then classified using the model sequentially to produce the complete segmentation map. A multi-scale pyramid consists of 3 layers: a crop of the superpixel, some context surrounding the crop, and the full image. In practice, we classify batches of superpixels. [*] This is a summary of the actual prompt. See full prompt in Appendix B.

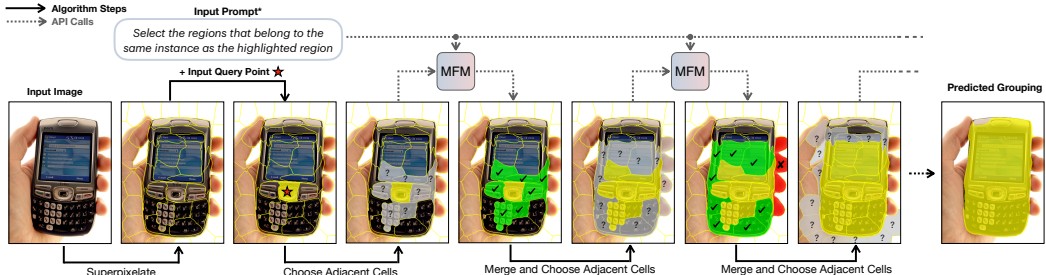

Figure 4: **Grouping algorithm.** Given an image and a query point, we first divide the image into superpixels and select the superpixel that the query point falls into. At each step, the model is asked to identify the adjacent superpixels that belong to the same object as the one covered by the cluster. The selected superpixels are then merged with the cluster to form the next step's input cluster. [*]This is a summary of the actual prompt. See full prompt in Appendix B.

*unsupervised* superpixel clustering algorithm (Achanta et al., 2012) and assign a single label per group to decrease the number of API calls (or forward passes). Using superpixels is a common approach to segmenting an image into smaller, homogeneous regions based on low-level image features, such as color or texture Stutz et al. (2018). We include calibration baselines to control the impact of the superpixelation (and other approximations in prompting) in Sec. 4.

After dividing the image into superpixels, we classify them in batches to decrease the overall cost as in the classification task. Similar to the object detection algorithm, this approach utilizes the strength of MFMs as good image classification models. To maintain consistency across different batches of superpixels, we include predictions for the previously obtained batches as part of the chain, which we found to improve the models' performance.

In our early experiments, we found that naively highlighting separate superpixels on an input image leads to poor performance. This is in line with other works (Fu et al., 2024; Wu & Xie, 2024) that found that MFMs have a "blurry vision" and struggle with fine-grained details and localization. To address this, we provide the MFM with the crops of each superpixel at multiple scales, which we found to improve the performance significantly. See Fig. 3 for overview and Alg. 3 for pseudo-code.

**Grouping**. Given an image and a query (or anchor) point on it, the grouping task consists of identifying other pixels that belong to the same object or background. Unlike semantic segmentation,

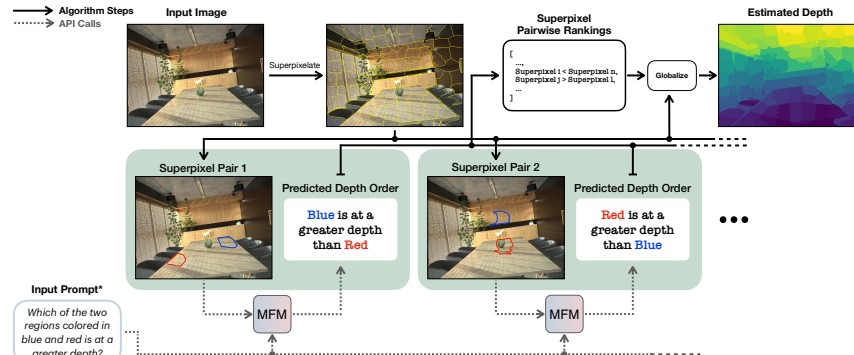

Figure 5: **Depth prediction algorithm**. We randomly select pairs of superpixels. Each pair is given to the model to perform a pairwise depth comparison. The resulting pairwise ranks are then globalized by minimizing an objective function to generate a relative depth map, which can then be scaled to obtain classical evaluation metrics. *This is a summary of the actual prompt. See full prompt in Appendix B.

there is no fixed, pre-defined set of classes, which makes it more challenging. As before, we make use of superpixels and the MFM's capability at determining visual similarity (Fu et al., 2024). We construct a graph where each superpixel is a node, and edges connect neighboring superpixels. We then identify the superpixel containing the query point and explore adjacent superpixels. The model decides whether each adjacent superpixel belongs to the same object as the initial superpixel. The selected superpixels are then merged with the initial one to form the next input cluster. This process continues until no more superpixels are added. See Fig. 4 for overview and Alg. 4 for pseudo-code.

**Depth prediction.** As predicting 3D from a single 2D image is inherently ambiguous, we perform relative depth prediction by querying the model to rank different parts of the image according to their distance from the camera. Like segmentation, querying at the pixel level directly from the image is infeasible. Instead, we adopt a region-wise comparison strategy similar to Zoran et al. (2015). To identify suitable regions for comparison, we first segment the image into superpixels. We then randomly sample pairs of superpixels and query the MFM to rank these pairs based on relative depth. These pairwise rankings are then globalized by minimizing the objective function from (Zoran et al., 2015), which encourages assigning larger values to superpixels ranked deeper than those ranked shallower in the pairwise comparisons (see C.3 for details). We then use the values assigned by the objective to rank all superpixels. For simplicity, we assume that all pixels within a superpixel share the same depth rank, allowing us to extend the superpixel-level depth predictions to a pixel-wise ranking across the entire image (control baselines are included in evaluations). Please see Fig. 5 for an overview and Algorithm 5 for pseudo-code.

**Surface normal prediction.** We follow a similar ranking approach as for depth. We use standard basis vectors relative to the camera (right, up, and forward) as reference directions, and for each randomly sampled pair of superpixels, we query the MFM to determine their relative alignment with each basis vector. After we obtain the pairwise comparisons for each direction, we globalize them using the same algorithm used for depth (Zoran et al., 2015). This results in three distinct surface normal maps, one for each basis direction. Similar to depth, we assume uniformity within superpixels and assign the same rank to all pixels within each superpixel group (control baselines are included in evaluations). Please see Fig. 6 for an overview and Algorithm 6 for pseudo-code.

# 4 EXPERIMENTS

In this section, we provide the experimental results for different tasks and MFMs. First, we describe our setting, including the choice of the datasets and models. Then, we discuss our main results. We provide qualitative examples for all tasks in Fig. 7. Finally, we provide further analysis and ablations in Sec. 4.1. Please see the Appendix Sec. A and E for additional results.

Figure 6: **Surface normal prediction algorithm.** Similar to the depth estimation algorithm, we randomly select superpixels and give them to the model to perform a pairwise comparison. The superpixels are compared based on their alignment with the basis vectors relative to the camera. The pairwise ranks are globalized to create a relative surface normal map. *This is a summary of the actual prompt. See full prompt in Appendix B.

**Tested Multimodal Foundation Models.** We perform evaluations of several closed-weight MFMs, namely GPT-4o (OpenAI, 2024), Gemini 1.5 Pro (Reid et al., 2024), and Claude 3.5 Sonnet (Anthropic, 2024) by querying them via their APIs. We also include Qwen2-VL-72B (Wang et al., 2024) as a recent open-weight model that was shown to be competitive with GPT-4o and Claude 3.5 Sonnet on some benchmarks. For each model and task, we first choose the best prompt out of several candidates based on a small validation set and use it to obtain the final results on a test set.

**Datasets.** In our evaluations, we use the following commonly employed vision datasets:

- **Image classification.** We use standard benchmarks including ImageNet (Russakovsky et al., 2014) and ImageNet-v2 (Recht et al., 2019). To test robustness, we include ImageNet-R (Hendrycks et al., 2021), ImageNet-S (Wang et al., 2019), and two corruption benchmarks from RobustBench (Croce et al., 2020), specifically, ImageNet-C (Hendrycks & Dietterich, 2019) and ImageNet-3DCC (Kar et al., 2022b).

- **Object detection.** We use the COCO (Lin et al., 2014) validation and choose images containing only a single instance of each present class, resulting in 1.7K examples.

- **Semantic segmentation & grouping.** We use a random subset of 500 COCO (Lin et al., 2014) validation images for semantic segmentation for cost-efficiency. For grouping, we filter 100 images from the COCO validation set by measuring the consistency of SAM (Kirillov et al., 2023a) predictions between different query points within every instances. More details are provided in Appendix E.3.

- **Depth & surface normal prediction.** We use Hypersim (Roberts et al., 2021) and randomly subsample 100 validation images from it for cost-efficiency.

**Baselines.** We include the following control baselines to judge the performance of MFMs:

- **Vision Specialist.** We report the performance of leading computer vision models for each task. We specify each model used in the corresponding task sections. This baseline indicates the current state of the (specialized) computer vision models.

- **Oracle + Chain.** This baseline shows the performance of the prompt chain if the MFM gave the ground-truth answer at each classification sub-task. This allows us to isolate the performance of MFMs from the limitations of the prompt chaining algorithm.

- **Vision Specialist + Chain.** This baseline applies the same algorithmic constraints to the vision specialist as those experienced by MFMs, such as superpixels and recursive zooming. This control baseline provides a fair, calibrated comparison between vision specialists and MFMs.

- **Blind Guess.** We prompt the model with a blank image, revealing potential biases and assessing whether the model genuinely utilizes the image content for its predictions.

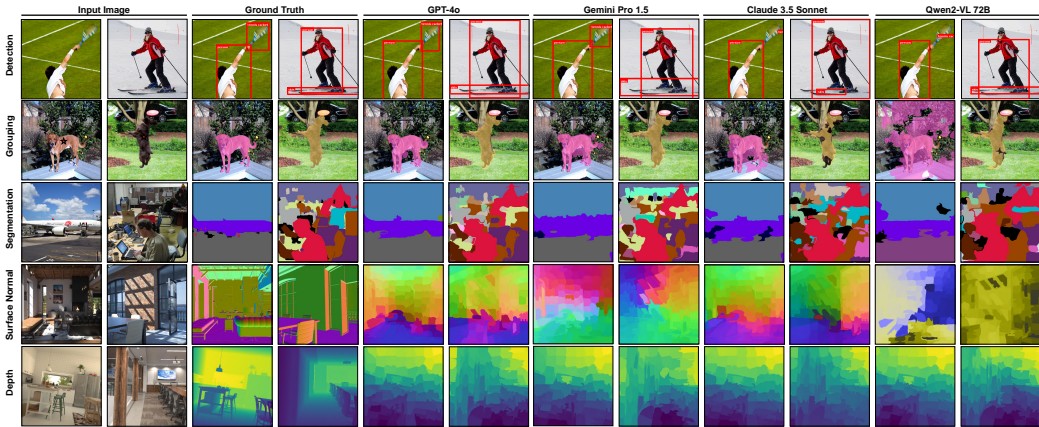

Figure 7: **Qualitative results.** Visual comparisons showing the performance of MFMs across each task. We find that all models perform relatively better on semantic tasks compared to the geometric ones. For surface normal visualizations, we combine the per-axis normalized predictions and project onto the unit sphere, see Appendix C.4 for details and Fig. 8 for more qualitatives.

Table 1: **Image classification.** We compare the performance of the MFMs with vision specialists, Model Soups (Wortsman et al., 2022) and OpenCLIP Cherti et al. (2023). Although their performance falls short of the top specialist models, MFMs, particularly GPT-4o, demonstrate competitive results across a broad range of benchmarks.

| Model | ImageNet | ImageNet-V2 | Corruptions | | Domain Shift | |
| --- | --- | --- | --- | --- | --- | --- |
| | | | (2DCC) | (3DCC) | (ImageNet-R) | (ImageNet Sketch) |
| Model Soups ViT-G | 90.94 | 84.22 | - | - | 95.46 | 74.23 |
| OpenCLIP H | 84.37 | 78.33 | 66.96 | 65.95 | 93.76 | 73.24 |
| GPT-4o | 77.196 | 71.57 | 62.46 | 61.13 | 84.38 | 67.30 |
| Gemini 1.5 Pro | 73.88 | 69.76 | 56.14 | 56.22 | 71.42 | 57.15 |
| Claude 3.5 Sonnet | 62.846 | 54.45 | 40.76 | 41.41 | 70.36 | 57.42 |
| Qwen2-VL | 55.54 | 49.39 | 38.92 | 36.45 | 66.31 | 51.18 |

**Image classification.** The classification results across all datasets are summarized in Tab. 1. We use Model Soups ViT-G (Wortsman et al., 2022) as the vision specialist, and we also include OpenCLIP H (Cherti et al., 2023) to assess zero-shot capabilities. Although MFMs do not reach the performance levels of vision specialists, they demonstrate strong results across the benchmarks and show resilience to image corruptions and natural distribution shifts. Notably, GPT-4o stands out with a particularly strong performance followed by Gemini 1.5 Pro, Claude 3.5 Sonnet, and Qwen2-VL.

**Object detection.** The results are summarized in Tab. 2. We use DETR (Carion et al., 2020), Co-DETR (Zong et al., 2023), a state-of-the-art COCO model, as the vision specialists. We observe that all MFMs lag behind the vision models, with GPT-4o achieving the highest performance, significantly outperforming other MFMs. The much lower $AP_{75}$ performance for the "chained" versions of the vision specialists suggests that the gap between them and MFMs can be partly explained by the restrictions of the chain algorithm (grid structure and zooming).

We also evaluate Gemini 1.5 Pro and Qwen2-VL by directly regressing the bounding boxes since they provide such capability, which was shown to be effective (Google, 2024). Interestingly, while their performance improves, we find that they still fall significantly behind the specialist models and still do not outperform GPT-4o with the chain algorithm.

Finally, we assess the performance of the "Oracle + Chain". Two baselines are evaluated: one using GPT-4o's class predictions, and another using the ground-truth class labels. The first baseline examines the outcome if GPT-4o correctly selects the grid cells at each step of the chain, while the second assumes both correct class predictions and accurate grid cell selection. These provide theoretical upper bounds for both the grid search component and the overall pipeline. The oracle baseline results for the other MFMs are provided in Appendix E.

Table 2: **Object Detection.** We compare the performance of MFMs against vision specialists, DETR (Carion et al., 2020) and Co-DETR (Zong et al., 2023). $^*$ indicate models evaluated with direct prompting to regress bounding boxes. Similar to the classification task, vision specialists significantly outperform MFMs, with GPT-4o achieving the best performance.

| Baselines | Model | $AP_{50}$ | $AP_{75}$ | AP |
|---|---|---|---|---|
| Vision Specialists | Co-DETR | 91.30 | 86.17 | 80.23 |
| | Co-DETR + Chain | 90.06 | 52.78 | 51.54 |
| | DETR | 73.31 | 63.61 | 58.67 |
| | DETR + Chain | 72.33 | 38.36 | 39.36 |
| MFMs | GPT-4o | 60.62 | 31.97 | 31.87 |
| | Gemini 1.5 Pro (direct)* | 55.11 | 31.23 | 31.33 |
| | Gemini 1.5 Pro | 39.75 | 15.27 | 18.11 |
| | Claude 3.5 Sonnet | 31.69 | 12.13 | 14.78 |
| | Qwen2-VL (direct)* | 44.10 | 23.71 | 24.36 |
| | Qwen2-VL | 35.62 | 12.82 | 15.27 |
| Control | Oracle + Chain (pred. class) | 75.44 | 41.31 | 41.56 |
| | Oracle + Chain (full) | 92.18 | 49.33 | 50.14 |
| | Blind guess | <0.01 | <0.01 | <0.01 |

Table 3: **Semantic Segmentation and Grouping.** We compare the performance of MFMs against OneFormer (Jain et al., 2022) and SAM (Kirillov et al., 2023b) vision specialist. Similar to classification and detection, all models show highly non-trivial performance in both tasks, with GPT-4o having a particularly strong performance, as can also be seen in qualitative results in Fig. 7.

Table 4: Semantic Segmentation Results.

| Baselines | Model | mIoU | Pixel Accuracy |
|---|---|---|---|
| Vision Specialists | OneFormer | 65.52 | 83.26 |
| | OneFormer + Chain | 60.64 | 81.69 |
| MFMs | GPT-4o | 40.50 | 65.03 |
| | Gemini 1.5 Pro | 36.90 | 60.20 |
| | Claude 3.5 Sonnet | 29.06 | 54.93 |
| | Qwen2-VL | 30.81 | 55.26 |
| Baselines | Oracle + Chain | 82.90 | 94.08 |
| | Blind guess | 0.5 | 8.34 |

Table 5: Grouping results.

| Models | mIoU |
|---|---|
| SAM | 80.12 |
| SAM + Chain | 72.32 |
| GPT-4o | 59.06 |
| Gemini 1.5 Pro | 44.13 |
| Claude 3.5 Sonnet | 41.68 |
| Qwen2-VL | 21.64 |
| Oracle + Chain | 81.77 |

**Semantic segmentation.** Tab. 4 and Fig. 7 show that MFMs achieve rather non-trivial performance, yet still significantly behind the vision specialist, i.e. OneFormer (Jain et al., 2022). Similar to object detection, we include the baseline of constraining the performance of the vision specialist using the chain algorithm: we assign the majority class prediction to each superpixel and flood-fill the entire superpixel with that class. We observe that the mIoU remains largely unaffected, suggesting that the constraints imposed by this approach are less stringent compared to the object detection.

**Grouping.** As an extension of the semantic segmentation task, we evaluate MFMs on a grouping task. Tab. 5 shows that MFMs have varying performance on this task, and GPT-4o performs the best, achieving overall good performance as can also be seen in Fig. 7,8. All models still lag behind the vision specialist SAM (Kirillov et al., 2023a).

**Depth prediction.** The results are summarized in Tab. 6. Alongside standard metrics, we also report **1)** The Spearman correlation coefficient ($\rho$), which serves as a relative metric by measuring the correlation between the ground-truth depth ranking of the pixels and the predicted ranking and **2)** Accuracy, which reflects the percentage of correct pairwise depth comparisons. While MFMs demonstrate non-trivial performance outperforming the blind guess, there remains a significant gap compared to the vision specialist, Omnidata (Kar et al., 2022a; Eftekhar et al., 2021). which is more pronounced compared to the semantic tasks. This suggest that their 3D understanding is worse than semantic one.

To assess the constraints imposed by the algorithm, consistent with our approach in previous tasks, we analyze the results when all queried pairwise comparisons are 100% accurate in the "Oracle + Chain" baseline, showing the upper bound performance that can be obtained from a limited set of pairwise comparisons. Finally, we restrict Omnidata to the constraints imposed by the algorithm

Table 6: **Depth prediction.** The numbers show that while the models exhibit a non-trivial ability to coarsely estimate depth from images, the gap is higher than for semantic tasks. Additionally, unlike the semantic tasks, the performance of all the MFMs is similar.

| Baselines | Method | Higher is better ↑ | | | | | Lower is better ↓ |
|---|---|---|---|---|---|---|---|
| | | $\delta_1$ | $\delta_2$ | $\delta_3$ | $\rho$ | Accuracy | AbsRel |
| Vision Specialists | Omnidata | 0.768 | 0.867 | 0.911 | 0.95 | - | 0.375 |
| | Omnidata + Chain | 0.568 | 0.772 | 0.864 | 0.81 | 93.74 | 0.528 |
| MFMs | GPT-4o | 0.459 | 0.712 | 0.838 | 0.53 | 70.59 | 0.621 |
| | Gemini 1.5 Pro | 0.458 | 0.709 | 0.835 | 0.51 | 66.78 | 0.628 |
| | Claude 3.5 Sonnet | 0.429 | 0.693 | 0.830 | 0.48 | 68.09 | 0.657 |
| | Qwen2-VL | 0.432 | 0.698 | 0.831 | 0.41 | 64.44 | 0.637 |
| Control | Oracle + Chain | 0.571 | 0.774 | 0.863 | 0.83 | 100.0 | 0.528 |
| | Blind Guess | 0.375 | 0.628 | 0.773 | 0.25 | 54.24 | 0.758 |

Table 7: **Surface normal prediction.** The numbers reveal that all the MFMs struggle with specific aspects of the task. They consistently confuse directions along the $x$ axis, a bias also observed in the blind guess baseline. Gemini, especially, exhibits significant difficulties, and shows a performance that is close to or even worse than random chance across all three directions.

| Baselines | Method | $\rho_x$ | $\rho_y$ | $\rho_z$ | Accuracy$_x$ | Accuracy$_y$ | Accuracy$_z$ |
|---|---|---|---|---|---|---|---|
| Vision Specialists | Omnidata | 0.78 | 0.83 | 0.80 | - | - | - |
| | Omnidata + Chain | 0.64 | 0.70 | 0.58 | 95.14 | 96.31 | 94.28 |
| MFMs | GPT-4o | -0.15 | 0.56 | 0.38 | 48.31 | 75.52 | 68.53 |
| | Gemini 1.5 Pro | -0.20 | -0.58 | 0.03 | 43.71 | 41.24 | 51.62 |
| | Claude 3.5 Sonnet | -0.21 | 0.61 | 0.38 | 48.16 | 77.61 | 66.95 |
| | Qwen2-VL | 0.10 | -0.08 | 0.02 | 50.17 | 47.25 | 50.07 |
| Control | Oracle + Chain | 0.64 | 0.70 | 0.60 | 100.0 | 100.0 | 100.0 |
| | Blind guess | -0.48 | -0.61 | 0.11 | 39.70 | 38.52 | 53.64 |

in the "Omnidata + Chain" baseline. The numbers reveal that this setup performs similarly to the oracle, indicating that MFMs would need to achieve near-perfect pairwise comparisons to reach comparable results.

**Surface normal prediction.** We employ two metrics to assess performance: **1)** Spearman's rank correlation coefficient, $\rho_i$, measuring the correlation between ground truth and predicted pixel alignments along each basis direction $i$. Alignment for a pixel is measured as the dot product of the surface normal with the direction $i$. **2)** The accuracy, Accuracy$_i$, of pairwise alignment rankings between superpixels along direction $i$.

Tab. 7 demonstrates that the MFMs struggle with the task: all models fail to achieve positive correlation along the left-right direction, with Gemini 1.5 Pro performing below random chance for all three directional components, revealing a consistent bias in its understanding of these directions. We show in Appendix D that this trend extends to other MFMs when using direct prompting; but the up-down ambiguity is resolved with chain-of-thought prompting. Similar to depth estimation, these results suggest that MFMs have poor 3D visual understanding.

## 4.1 ANALYSIS AND ABLATIONS

**Prompt chaining vs naive prompting.** We analyze the impact of using the prompt chain and naive prompting in Tab. 8. Specifically, for bounding box regression, we directly query GPT-4o for coordinates, while for semantic segmentation, we mark image regions and request corresponding semantic labels. The results indicate a clear performance boost from using the prompt chain. We refer the reader to Appendix E for a detailed discussion, qualitative visuals, and other ablations.

**Prompt sensitivity.** We evaluate the MFMs across various prompts to assess their sensitivity to word choice and prompt structure. We then select the most effective prompt on a small validation set for the final results presented in Sec. 4. A comprehensive analysis is provided in Appendix D, showing that there is some variation in performance with different prompts, and the performance is generally less prompt-dependent for better-performing ones, e.g., GPT-4o.

Table 8: **Prompt chaining ablation.** We compare the performance of the prompt chaining algorithm with naive prompting techniques on GPT-4o for the tasks of semantic segmentation and object detection. As demonstrated, through prompt chaining the model's performance on both tasks enhanced significantly. * Segmentation is performed on a subset of 100 images.

| Task | Naive | Prompt Chaining |
|---|---|---|
| Segmentation (mIoU)* | 24.50 | **40.50** |
| Object Detection ($AP_{50}$) | 11.83 | **60.62** |

**In-the-wild evaluations.** Previously, we used standard vision datasets like ImageNet and COCO in our evaluations, which the model could have seen during pre-training. To assess whether the MFMs generalize to entirely novel data, we curated a collection of images released online within the past month (Flickr, 2024; Unsplash, 2024), which the MFMs could not have encountered during training. The results in Appendix E.7 show good generalization performance to the in-the-wild samples.

**Cost analysis.** A key consideration is the cost associated with prompting the MFMs; therefore, we provide detailed prompting costs for the prompt chains in Appendix F.

## 5 LIMITATIONS AND CONCLUSIONS

We investigate the vision capabilities of MFMs by translating standard computer vision tasks into an API-compatible format that can be solvable via prompt chaining. Our results show that the MFMs have relatively stronger performance in semantic tasks compared to geometric tasks, and GPT-4o is generally the best performing model, followed by Gemini 1.5 Pro, Claude 3.5 Sonnet, and Qwen2-VL-72B. All MFMs lag significantly behind the vision specialists for all tasks, suggesting plenty of room for improvement in model development. Furthermore, while we improved "myopic" visual perception of current MFMs (Rahmanzadehgervi et al., 2024) by incorporating different techniques in prompting such as zooming and super-pixelation, more work is needed on the prompting aspect to further reduce this tendency. Below we discuss some limitations of our work and some future directions.

*Extension to other vision tasks.* Our results and control baselines show that there exist relatively convenient prompt chains that decompose vision tasks into different sub-tasks that are solvable by MFMs. Developing similar chaining strategies for other vision tasks such as optical flow, 3D semantics, etc. for a broader coverage is a fruitful future direction. We believe our exploration and code serve as a good starting point.

*Improved prompt chaining.* As shown in the qualitative and quantitative results, our prompting approaches unlocked vision capabilities of current MFMs for several tasks. We also controlled for the impact of those choices on the comparative studies using appropriate control baselines. However, there is room for improving the absolute performance via including stronger chain-of-thought strategies and pixel approximations. Another potentially interesting direction is incorporating in-context learning and alignment techniques for boosting the performance.

*Costs, scalability, and inference time.* The computational costs (provided in Appendix D) remain higher than vision specialists for the MFMs. These costs are expected to come down as MFM inference becomes cheaper. A promising direction for future work is to develop algorithms with improved "prompt complexity", optimizing the balance between performance and token consumption.

*Data contamination.* The issue of data contamination is a broad concern for the community Jacovi et al. (2023). While our evaluations show the conclusions are generalizable, the further development of evaluation sets could help achieve a more unbiased assessment of performance.

*Absolute metrics for geometry.* We utilize relative metrics for depth and surface normal prediction, acknowledging the inherent ambiguity in these tasks. Our initial attempts to recover metric depth from monocular images were unsuccessful. While this simplifies the evaluation, future research could focus on employing techniques like in-context learning to achieve metric depth estimates.

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

APPENDIX

TABLE OF CONTENTS

## A  QUALITATIVE EXAMPLES

We provide additional qualitatives in Figures 8 and 9 to show each model's performance on different tasks.

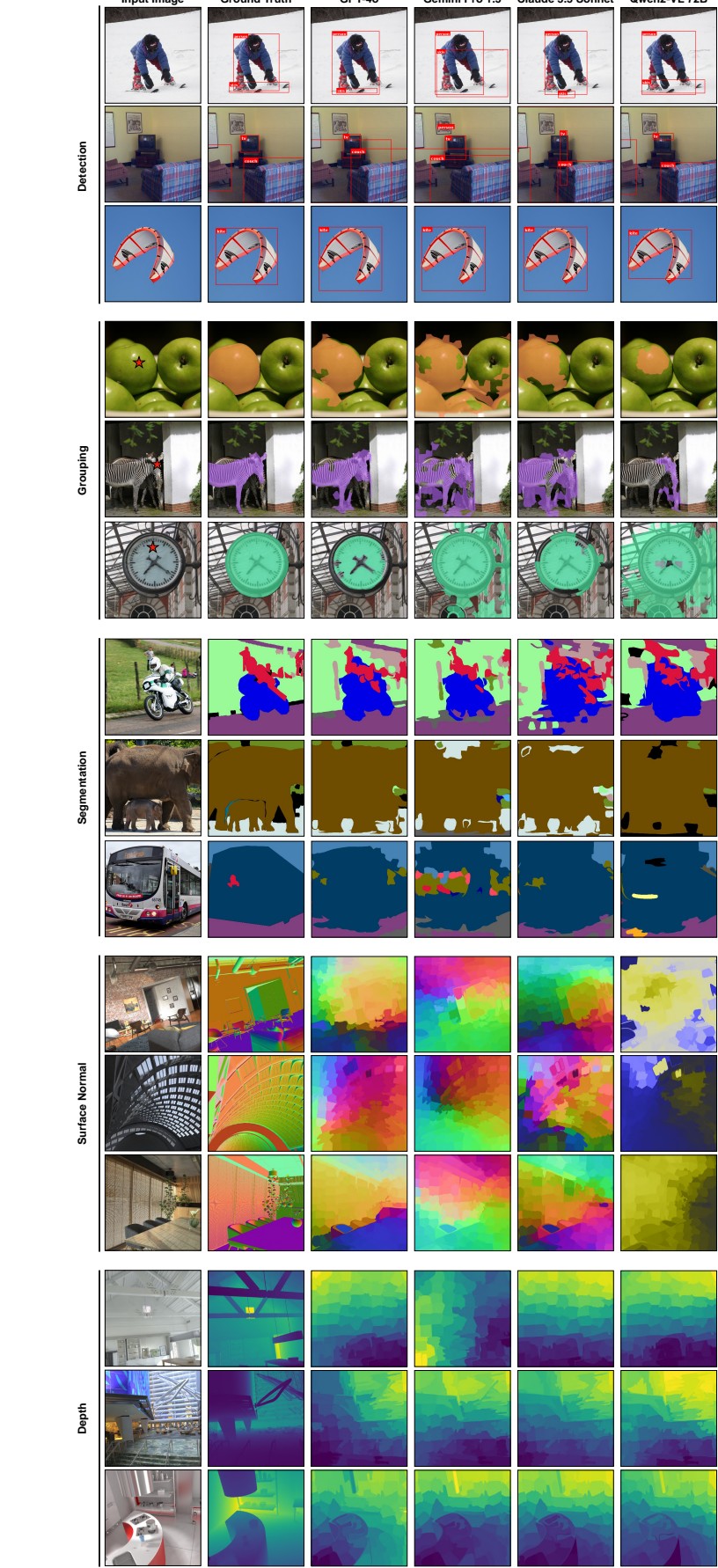

Figure 8: Additional qualitative results for MFM predictions on different tasks.

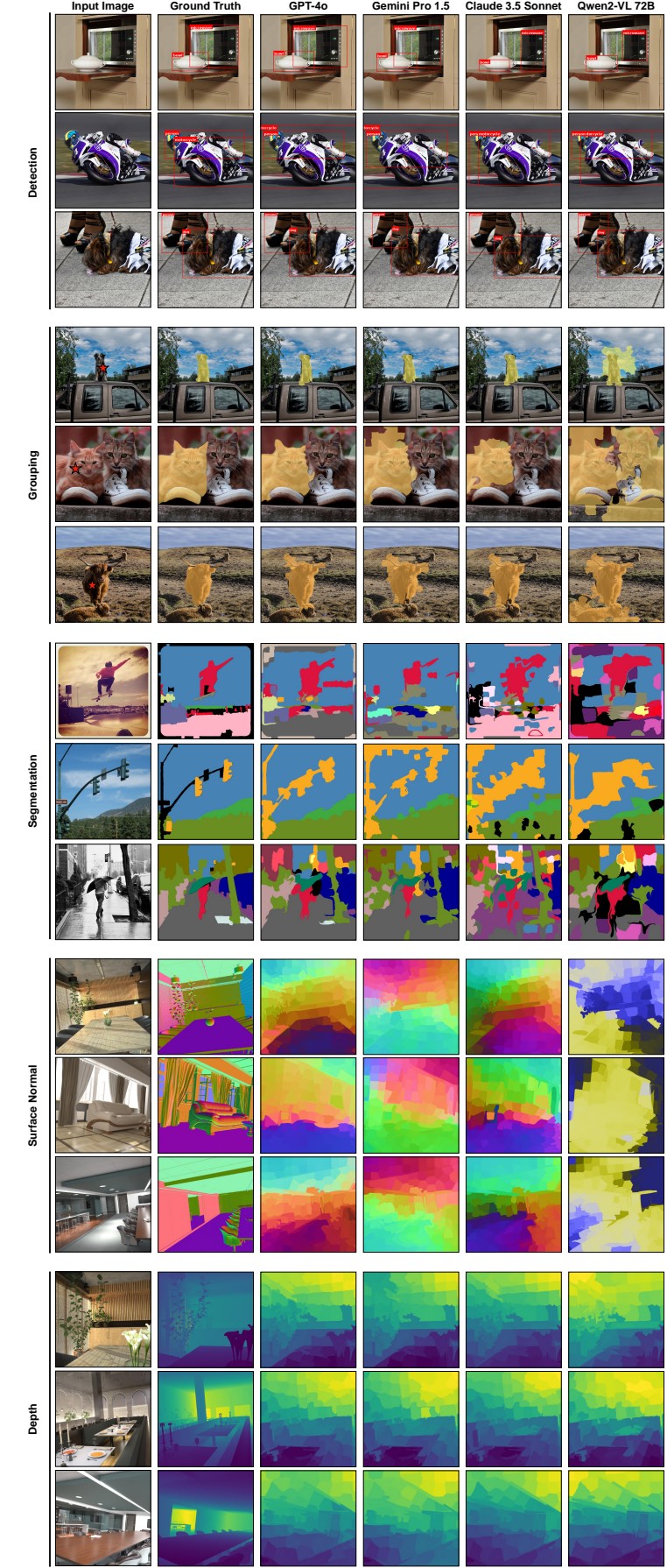

Figure 9: Additional qualitative results for MFM predictions on different tasks.

## B   FULL PROMPTS

We provide full prompts in the supplementary material.

## C   ADDITIONAL DETAILS ON PROMPT CHAINING

### C.1   OBJECT DETECTION

**Different variations of classification for object detection.** As discussed in Section 3, the first stage of the object detection pipeline involves identifying all the objects present in the image. We attempt the following two strategies for the multi-label classification task:

- The first strategy simply provides the model with the entire image, asking it to identify all present classes.

- The second strategy divides the image into five regions: four quadrants and a center crop. The model is asked to identify the classes present in the 5 regions in independent queries. With each query, the full image is provided for additional context. The final prediction is obtained by taking the union of the classes identified across all regions (see Algorithm 1 in the appendix for detailed pseudocode). This approach typically improves recall but may reduce precision, reflecting a trade-off between the two strategies.

The precision-recall trade-off for the models is described in Tab. 9. To pick the best classification strategy for the models, we run the oracle on the predicted labels on a small subset and pick the one that yields the highest AP.

After we find the object labels, we run the procedure described in Algorithm 2 to regress the bounding boxes.

---

**Algorithm 1** Region-based Image Classification

---

1: **procedure** REGIONBASEDCLASSIFICATION($image$)
2:     $regions \leftarrow$ DivideIntoRegions($image$)
3:     $allClasses \leftarrow \emptyset$
4:     **for** $region \in regions$ **do**
5:         $classes \leftarrow$ QueryMFM($image, region$)
6:         $allClasses \leftarrow allClasses \cup classes$
7:     **end for**
8:     **return** $allClasses$
9: **end procedure**
10: **procedure** DIVIDEINTOREGIONS($image$)
11:     $quadrants \leftarrow$ DivideIntoQuadrants($image$)
12:     $center \leftarrow$ ExtractCenterCrop($image$)
13:     **return** $quadrants \cup \{center\}$
14: **end procedure**

---

Table 9: **Classification for Object Detection:** The results clearly show the precision-recall trade-off between using the two strategies for multi-label classification.

| Strategy | Model | Precision | Recall |
|----------|-------|-----------|--------|
| Strategy 1 | GPT-4o | 97.5 | 75.75 |
| | Gemini 1.5 Pro | 90.5 | 83.81 |
| | Claude 3.5 Sonnet | 84.27 | 81.24 |
| Strategy 2 | GPT-4o | 89.05 | 88.37 |
| | Gemini 1.5 Pro | 84.37 | 89.3 |
| | Claude 3.5 Sonnet | 78.18 | 85.94 |

---

**Algorithm 2** Recursive Grid-Search

---

1: **procedure** COARSEGRIDSEARCH($image, object, gridStructure$)
2:     **while** search space can be reduced **do**
3:         $cells \leftarrow$ DivideIntoGrid($image, gridStructure$)
4:         $relevantCells \leftarrow \{c \in cells : $QueryMFM$(c, object) = $TRUE$\}$
5:         $image \leftarrow$ CropToRelevantCells($image, relevantCells$)
6:     **end while**
7:     **return** $image$ as $bbox$
8: **end procedure**
9: **procedure** QUERYMFM($cell, object$)
10:     **return** MFM classification of object presence in cell
11: **end procedure**

---

### C.2 SEGMENTATION

The procedures for supervised segmentation and grouping are described in Algorithm 3 and Algorithm 4 respectively.

---

**Algorithm 3** Superpixel Segmentation

---

1: **procedure** SEMANTICSEGMENTATION($image, batchSize, scaleList$)
2:     $superpixels \leftarrow$ SLIC($image$)
3:     $classifiedSuperpixels \leftarrow \emptyset$
4:     $history \leftarrow \emptyset$
5:     **for** $i \leftarrow 1$ to length($superpixels$) step $batchSize$ **do**
6:         $batch \leftarrow$ GetBatch($superpixels, i, batchSize$)
7:         $semanticPyramid \leftarrow$ CreateSemanticPyramid($image, batch, scaleList$)
8:         $batchClasses \leftarrow$ ClassifyBatch($semanticPyramid, history$)
9:         $classifiedSuperpixels \leftarrow classifiedSuperpixels \cup batchClasses$
10:         $history \leftarrow$ UpdateHistory($history, batchClasses$)
11:     **end for**
12:     $segmentedImage \leftarrow$ FloodFillSuperpixels($image, classifiedSuperpixels$)
13:     **return** $segmentedImage$
14: **end procedure**

---

**Algorithm 4** BFS Segmentation

---

1: **procedure** UNSUPERVISEDSEGMENTATION($image, queryPoint, batchSize, scaleList$)
2:     $superpixels \leftarrow$ SLIC($image$)
3:     $graph \leftarrow$ ConstructSuperpixelGraph($superpixels$)
4:     $startNode \leftarrow$ FindSuperpixelContaining($superpixels, queryPoint$)
5:     $cluster \leftarrow \{startNode\}$
6:     $queue \leftarrow$ new Queue()
7:     $queue$.enqueue($startNode$)
8:     $visited \leftarrow \{startNode\}$
9:     **while** not $queue$.isEmpty() **do**
10:         $batch \leftarrow$ GetBatchFromQueue($queue, batchSize$)
11:         $batchPyramid \leftarrow$ CreateSemanticPyramid($image, batch, scaleList$)
12:         $clusterPyramid \leftarrow$ CreateSemanticPyramid($image, cluster, scaleList$)
13:         $newMembers \leftarrow$ QueryMFM($batchPyramid, clusterPyramid$)
14:         $cluster \leftarrow cluster \cup newMembers$
15:         $queue, visited \leftarrow$ UpdateQueueAndVisited($graph, newMembers, visited$)
16:     **end while**
17:     **return** $cluster$
18: **end procedure**

---

## C.3 DEPTH ESTIMATION

The procedure for depth estimation is given in Algorithm 5. A crucial part of the algorithm involves optimizing the objective to obtain the overall depth rankings. To formulate the objective for globalizing the pairwise depth rankings, we re-purpose the objective in Zoran et al. (2015). Given the vector of global rankings $\boldsymbol{x} \in \mathbb{R}^N$, we first consider instances where superpixel $i$ is predicted to be at a greater depth than superpixel $j$. The corresponding objective is formulated as:

$$\mathcal{L}_{gt}(\boldsymbol{x}) = \sum_{i,j} (x_i - x_j - 1)^2 \tag{1}$$

This objective encourages $x_i$, ranked at a greater depth than $x_j$, to take on higher values. Similarly, an analogous objective $\mathcal{L}_{lt}$ can be defined for superpixels $x_i$ predicted to be at a lesser depth than $x_j$.

Following Zoran et al. (2015), we include a smoothness regularization term to stabilize the depth estimations:

$$\mathcal{L}_s(\boldsymbol{x}) = \sum_{i,j} (x_i - x_j)^2 \tag{2}$$

This regularization is applied over pairs of adjacent superpixels $i$ and $j$, promoting smooth transitions between their depth values.

The final objective that needs to be minimized is a weighted sum of the above terms:

$$\boldsymbol{x} = \min_{\boldsymbol{x}} \left( \lambda_{gt} \mathcal{L}_{gt} + \lambda_{lt} \mathcal{L}_{lt} + \lambda_s \mathcal{L}_s \right) \tag{3}$$

where $\lambda_{gt}$, $\lambda_{lt}$, and $\lambda_s$ are the weight parameters. For our experiments, we select $\lambda_{gt} = \lambda_{lt} = 1$ and $\lambda_s = 20$.

To obtain metric depth estimates, we assume access to ground-truth depth values for the purpose of scaling. Specifically, after flood-filling the values of $\boldsymbol{x}$, we generate a complete relative depth map $\boldsymbol{d}$. Given the ground-truth depth map $\boldsymbol{d}^*$, we optimize the following objective to determine the appropriate scale and shift parameters:

$$(s, t) = \arg\min_{s,t} \sum_{i=1}^{M} (s\boldsymbol{d}_i + t - \boldsymbol{d}_i^*)^2 \tag{4}$$

where $M$ is the total number of pixels in the image. By solving this optimization problem, we can then scale and shift the relative depth map $\boldsymbol{d}$ to align it with the metric depth.

---

**Algorithm 5** Depth Estimation

---

 1: **procedure** ESTIMATEDEPTH($image, numPairs$)
 2:     $superpixels \leftarrow$ SLIC($image$)
 3:     $pairwiseRankings \leftarrow \emptyset$
 4:     **for** $i \leftarrow 1$ to $numPairs$ **do**
 5:         $pair \leftarrow$ SampleRandomPair($superpixels$)
 6:         $ranking \leftarrow$ QueryMFM($pair$)
 7:         $pairwiseRankings \leftarrow pairwiseRankings \cup \{ranking\}$
 8:     **end for**
 9:     $globalRankings \leftarrow$ MinimizeObjective($pairwiseRankings$)
10:     $depthMap \leftarrow$ AssignDepthToPixels($image, superpixels, globalRankings$)
11:     **return** $depthMap$
12: **end procedure**

---

## C.4 SURFACE NORMAL ESTIMATION

The procedure for surface normal estimation is detailed in Algorithm 6. While the model makes binary decisions regarding whether one depth is lesser or greater than another, we have found that enabling the model to also consider equality predictions enhances the accuracy of surface normal estimations.

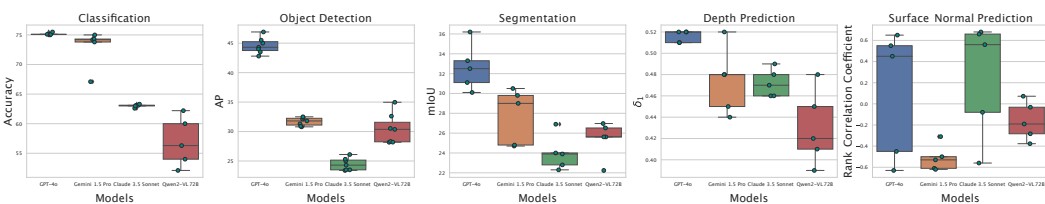

Figure 10: Sensitivity of MFMs to different prompting techniques. We observe that GPT-4o show-cases a lower sensitivity on most tasks compared to other MFMs.

To incorporate this into our approach, we introduce the following term for cases where superpixels $x_i$ and $x_j$ are predicted to be at equal depth:

$$\mathcal{L}_{eq}(\boldsymbol{x}) = \sum_{i,j}(x_i - x_j)^2 \tag{5}$$

for pairs of superpixels $x_i$ and $x_j$ predicted to lie at an equal depth. For the weights, we choose $\lambda_{eq} = \lambda_{lt} = \lambda_{gt} = 1$ and $\lambda_s = 20$.

---

**Algorithm 6** Surface Normal Estimation

---

1: **procedure** ESTIMATESURFACENORMALS($image, numPairs, bases$)
2:     $superpixels \leftarrow$ SLIC($image$)
3:     $pairwiseAlign \leftarrow \{\}$
4:     **for** $i \leftarrow 1$ to $numPairs$ **do**
5:         $pair \leftarrow$ SampleRandomPair($superpixels$)
6:         **for** $basis$ in $bases$ **do**
7:             $alignment \leftarrow$ QueryMFM($pair, basis$)
8:             $pairwiseAlign[basis] \leftarrow pairwiseAlign[basis] \cup \{alignment\}$
9:         **end for**
10:     **end for**
11:     $normalMaps \leftarrow \{\}$
12:     **for** $basis$ in $bases$ **do**
13:         $globalAlign \leftarrow$ MinimizeGlobalObjective($pairwiseAlign[basis]$)
14:         $normalMaps[basis] \leftarrow$ AssignAlignmentToPixels($image, superpixels, globalAlign$)
15:     **end for**
16:     **return** $normalMaps$
17: **end procedure**

---

To visualize surface normals, we take the per-axis predictions and normalize them to [0,1], after which we project them onto the unit sphere. We directly interpret the three channels as RGB values. Note that since the per-axis normalized surface normal predictions do not present absolute directional information with respect to the camera, the colors might not match the ground truth visualizations.

## D   PROMPT SENSITIVITY ANALYSIS

In Fig. 10 we evaluate the models for each task considering different prompting techniques. We observe that GPT-4o generally shows lower sensitivity to different prompts on most of the tasks compared to other MFMs. For surface normals, we interestingly observe that the predictions greatly improve in the $y$ and $z$ directions, when GPT-4o and Claude are asked to reason in the prompt.

## E   ADDITIONAL EXPERIMENTAL DETAILS AND RESULTS

### E.1   OBJECT DETECTION

We evaluate additional baselines for GPT-4o in Tab. 10. In these experiments, the classification component of the pipeline remains unchanged, while the grid search is replaced with alternative

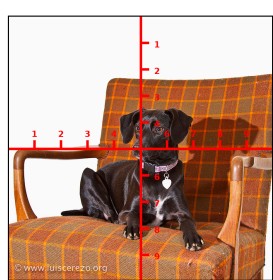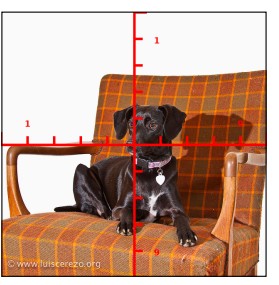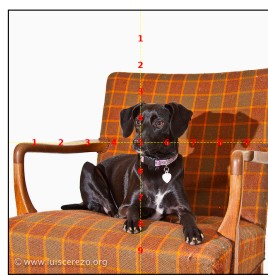

Figure 11: Different ruler types attempted as visual aids for object detection.

methods. The results are clear: GPT-4o struggles with directly regressing bounding box coordinates. To address this, we experimented with overlaying rulers on the images to assist in bounding box regression, following insights from Wu et al. (2024), but we found minimal improvement. The various visual prompts we tried are displayed in Fig. 11, and the numbers we obtained on a subset of 100 COCO images are summarised in Tab. 11

Table 10: **Additional experiments with MFMs on object detection.** Direct bounding box regression is ineffective for GPT-4o and Claude 3.5 Sonnet, while Gemini 1.5 Pro and Qwen2-VL perform better.

| Method | $AP_{50}$ | $AP_{75}$ | AP |
|---|---|---|---|
| GPT-4o (Direct Regression) | 11.83 | 1.33 | 3.24 |
| Gemini 1.5 Pro (Direct Regression) | 55.11 | 31.23 | 31.33 |
| Claude 3.5 Sonnet (Direct Regression) | 15.57 | 2.32 | 5.06 |
| Qwen2-VL (Direct Regression) | 44.10 | 23.71 | 24.36 |
| GPT-4o (Regression with Ruler) | 15.95 | 2.60 | 4.99 |

### E.2 SEMANTIC SEGMENTATION

We depict various marker types used for segmentation in Fig. 12. Furthermore, we conduct an ablation study on the marker type and the context provided during classification, as shown in Tab. 12. The numbers highlight the importance of contextual information within the semantic pyramid. Removing the context layer leads to a performance drop of over 10 mIoU. Additionally, the naive strategy of marking directly on the image and then classifying results in a 16 mIoU difference, indicating that MFMs currently lack the ability to localize precisely. We also investigate the impact of omitting the finest level of the semantic pyramid—the crop. While the mIoU value does not decrease much, qualitative analysis reveals that this omission hampers the model's ability to capture finer image details. This is shown in Fig. 13.

We also conduct ablation studies on the effect of the model's performance when the semantic pyramid is omitted. The visual markers in Fig. 12 don't work well for this, so we borrow a visual marker similar to the one used in Yang et al. (2023a) (see Fig. 14). Tab. 13 shows the results when the curve marker is used Yang et al. (2023a). It is clear that the model's performance greatly drops when it

Table 11: **Rulers for Object Detection:** The results indicate that visual markers such as rulers are ineffective in aiding GPT-4o for bounding box regression. Numbers obtained are on a subset of 100 COCO Images.

| Visual Prompt | $AP_{50}$ | $AP_{75}$ | AP |
|---|---|---|---|
| Ruler 1 | 21.19 | 4.09 | 7.60 |
| Ruler 2 | 22.59 | 7.85 | 9.20 |
| Ruler 3 | 19.06 | 4.86 | 8.09 |

**Point marker**   **Rectangle marker**   **Curve marker**

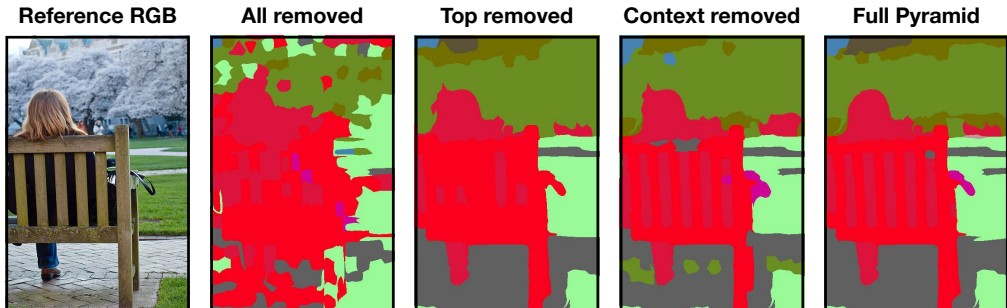

Figure 12: The curve, rectangle, and point marker types were tried for segmentation.

**Reference RGB**   **All removed**   **Top removed**   **Context removed**   **Full Pyramid**

Figure 13: **Semantic Segmentation predictions with different layers of the semantic pyramid**. From left to right: **1.** The RGB Image. **2.** The predicted mask when no crops are given, and markings on the full image are directly used. The model is unable to make out fine details. **3.** The predicted mask when the top of the semantic pyramid is removed. The model misses out on predicting some finer details (for instance, the gaps in the bench and the handbag). **4.** The predicted mask when the middle layer (the context) is removed. The model makes some wrong predictions. **5.** The mask with the full pyramid of information.

is deprived of the crops. We note that the marks we use differ from the ones used in Yang et al. (2023a) in two ways:

- The marks obtained in Yang et al. (2023a) already correspond to semantic entities, while we use superpixels as a proxy for this.

- Extracting a full semantic mask requires discerning finer-grained details, so the marks we use typically correspond to smaller regions in the image.

**Number markers**   **Rectangle markers**   **Curve markers**

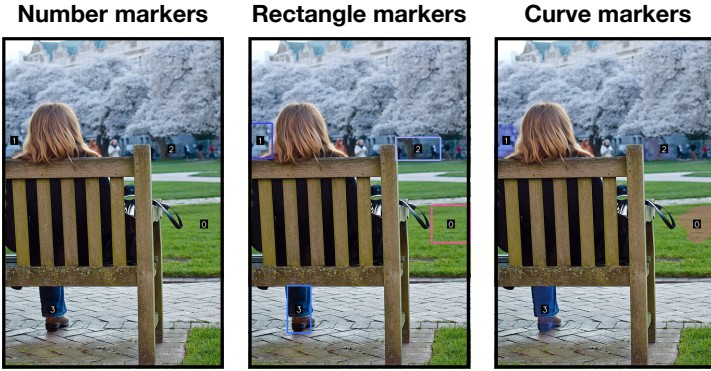

Figure 14: The marker styles used for directly querying semantic entities from the full image.

Table 12: **Ablation study on semantic segmentation.** The results show that GPT-4o is robust to the choice of visual prompt. The substantial performance drop (16 mIoU) observed upon removal of the semantic pyramid shows the critical role of the contextual information used in the sub-task.

| Category | Ablation | mIoU | Pixel Accuracy |
|---|---|---|---|
| Visual Prompts | Curve | 39.28 | 64.03 |
| | Rectangle | 40.52 | 65.99 |
| | Point | 40.10 | 65.19 |
| Contextual Ablations | Without Crop | 39.31 | 65.83 |
| | Without Context | 29.78 | 59.31 |
| | Best Naive | 24.50 | 51.01 |

Table 13: **Ablation on Direct Segmentation:** The numbers clearly show that omitting the extra information provided by the crops greatly impacts the model's performance. The numbers shown are for a subset of 30 images.

| Number of Superpixels | mIoU | Pixel Accuracy |
|---|---|---|
| 50 | 18.68 | 41.71 |
| 100 | 19.69 | 42.92 |
| 200 | 18.24 | 43.88 |
| 400 | 19.34 | 43.35 |

### E.3 GROUPING

For the grouping task, we filter out 100 COCO images that contain instances which are well-posed for this task. The well-posedness of an instance for grouping is measured by how consistent the SAM predictions are for the instance. To calculate the consistency of predictions for an instance, we sample random points inside the instance and use SAM to obtain an instance mask for each point individually, as well as a global mask by querying all points together. The mIoU between individual masks and the global mask is used as the consistency metric. Finally, the images that contain instances with a consistency value above a given threshold are selected and randomly sampled to create the evaluation set.

### E.4 DEPTH PREDICTION

We conduct an ablation study on the choice of visual markers in Tab. 14. Please also see Tab. 15 for additional oracle evaluations.

### E.5 SURFACE NORMAL PREDICTION

We conduct an ablation study on the choice of visual markers in Tab. 16.

Table 14: **Ablation study on depth estimation.** GPT-4o performs the best when curves are used as the visual marker.

| Method | Higher is better ↑ | | | | | Lower is better ↓ |
|---|---|---|---|---|---|---|
| | $\delta_1$ | $\delta_2$ | $\delta_3$ | $\rho$ | Accuracy | AbsRel |
| Curve | 0.550 | 0.822 | 0.935 | 53.75 | 70.43 | 0.332 |
| Rectangle | 0.534 | 0.807 | 0.931 | 51.68 | 69.28 | 0.341 |
| Point | 0.525 | 0.802 | 0.928 | 51.89 | 62.07 | 0.366 |

Table 15: **Oracle depth results** with different numbers of superpixels and comparisons made during chaining.

| Superpixels | Samples | Higher is better ↑ | | | | Lower is better ↓ |
|---|---|---|---|---|---|---|
| | | $\delta_1$ | $\delta_2$ | $\delta_3$ | $\rho$ | AbsRel |
| 100 | 200 | 0.571 | 0.774 | 0.863 | 0.83 | 0.528 |
| 100 | 400 | 0.597 | 0.785 | 0.867 | 0.86 | 0.514 |
| 200 | 200 | 0.571 | 0.773 | 0.867 | 0.83 | 0.501 |
| 200 | 400 | 0.593 | 0.788 | 0.869 | 0.86 | 0.502 |

Table 16: **Ablation study on surface normal estimation.** GPT-4o is relatively robust to different visual marker choices.

| Method | $\rho_x$ | $\rho_y$ | $\rho_z$ | $\text{Accuracy}_x$ | $\text{Accuracy}_y$ | $\text{Accuracy}_z$ |
|---|---|---|---|---|---|---|
| Curve | -4.89 | 58.00 | 39.28 | 49.02 | 67.95 | 66.9 |
| Rectangle | -13.99 | 58.84 | 39.65 | 45.75 | 69.25 | 67.83 |
| Point | 2.42 | 51.26 | 39.59 | 49.65 | 67.55 | 67.2 |

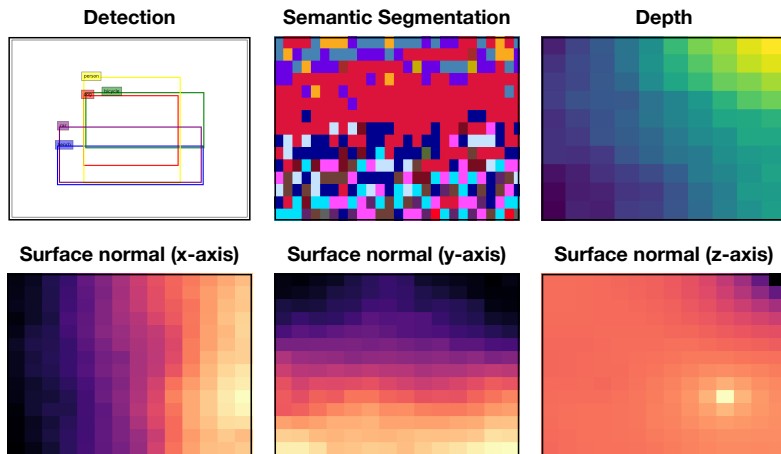

Figure 15: The blind guesses made by GPT-4o on different tasks.

### E.6 BLIND GUESS

As mentioned in Section 4, a useful way to analyze the potential biases of the MFM, and to gauge the degree to which it uses the visual content is a blind guess, or prompting the image with a blank image. In particular:

- For **object detection**, we ask the model to imagine classes present. After this, we ask it to provide reasonable coordinates for the objects based on its world knowledge.

- For **semantic segmentation**, we mark a rectangle in a white image and force the model to predict a class. We ask the model to use the location to make an educated guess.

- For **depth**, we ask the model to imagine an indoor setting. We mark two rectangles and force the model to predict that one is at a greater depth than the other.

- For **normals**, we repeat the procedure for depth for each direction.

The results for GPT-4o are visualized in Fig. 15, and reveal several interesting insights.

- For **object detection**, the model chooses common classes like person and car. Additionally, it seems to grasp the relative sizes of objects reasonably well, as indicated by its tendency to make the car and the bench longer.

- For **semantic segmentation**, the model makes reasonable guesses. For instance, it guesses "sky-merged" and "airplane" at the top of the image, "person" near the middle, "dog," "cat," and "floor" near the bottom.

- For **depth estimation**, GPT-4o exhibits a "ceiling bias" and consistently infers that the top right corner is located at a greater relative depth. We observe that this bias is reflected in several of the model's predictions as well, where the ceiling is consistently assumed to be at a greater depth.

- For **surface normals**, the model uses the relative locations of the rectangles to form judgments. For instance, in the $x$ direction, it infers that the right rectangle aligns more towards the right. In the $y$ direction, it consistently infers that the bounding box at a greater $y$ coordinate aligns more with the positive $y$ direction. While Chain-of-Thought (CoT) reasoning is able to break this bias along the $y$ direction for GPT-4o, the left-right bias persists when actual images are presented.

### E.7 IN-THE-WILD EVALUATIONS

Please see Figure 16 for qualitative evaluation of MFMs on in-the-wild samples.

## F PROMPTING COSTS

The costs for all the scaled-up experiments are documented in Tab. 17.

Table 17: Prompting costs for scaled-up experiments (in $)

| Task | GPT-4o | Gemini 1.5 Pro | Claude 3.5 Sonnet |
|---|---|---|---|
| Classification (ImageNet) | 47.54 | 63.87 | 30.94 |
| Object Detection | 185.76 | 610.83 | 155.04 |
| Semantic Segmentation | 232.07 | 450.14 | 227.87 |
| Grouping | 22.31 | 47.41 | 42.03 |
| Depth | 57.35 | 52.36 | 198.17 |
| Normals | 130.11 | 50.05 | 209.92 |

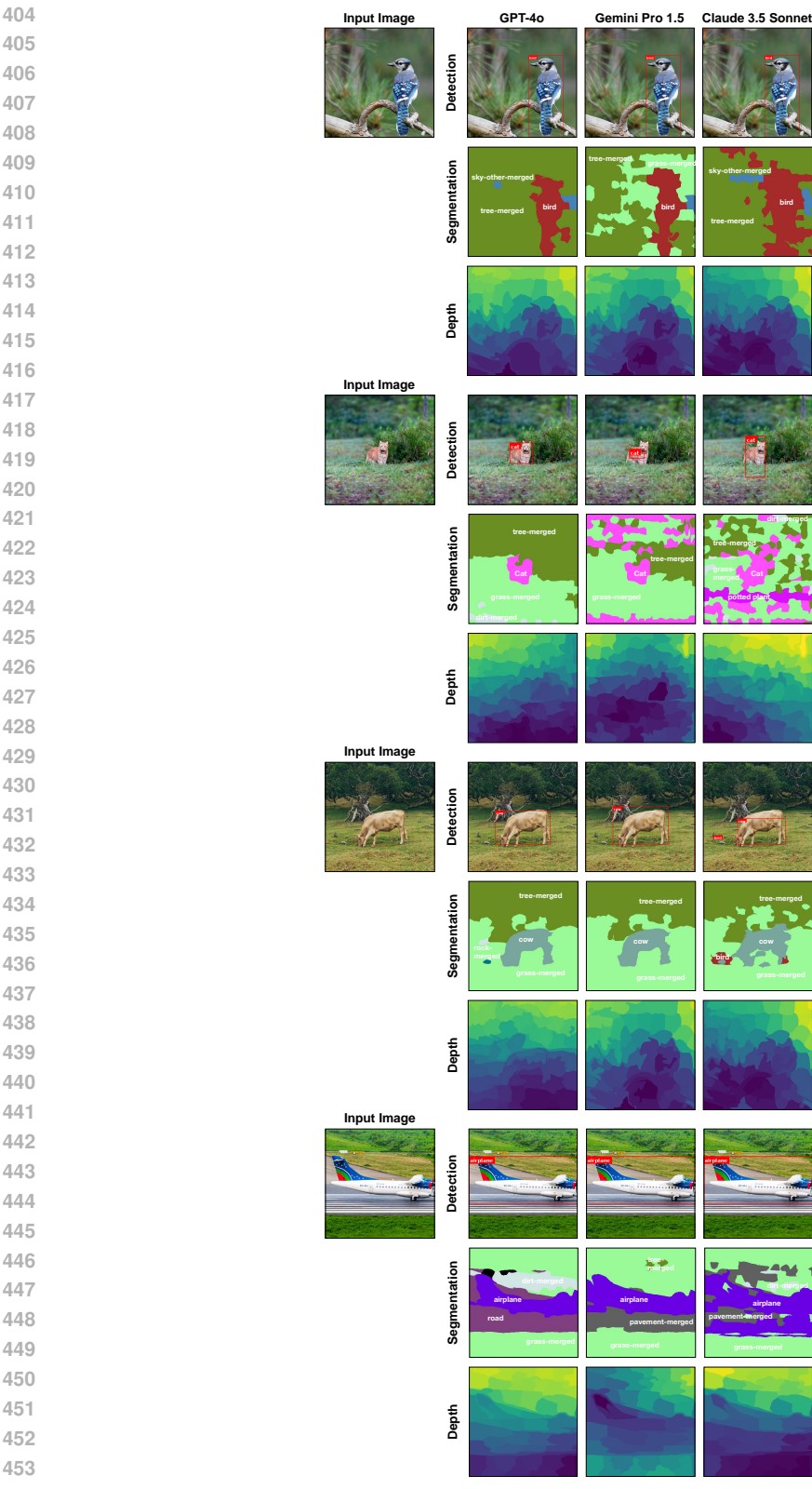

Figure 16: Qualitative results of evaluating MFMs on in-the-wild examples.

