# OpenReview forum: "How well does GPT-4o understand vision? Solving standard computer vision tasks with multimodal foundation models"
_ICLR.cc/2025/Conference — ICLR 2025 Conference Withdrawn Submission_

### Official Review · Reviewer_FUEP · 2024-10-19

**Soundness:** 3
**Presentation:** 3
**Contribution:** 3
**Rating:** 5
**Confidence:** 5

**Summary:**

This paper evaluates the performance of popular multimodal foundation models (MFMs), such as GPT-4o, Gemini 1.5 Pro, Claude 3.5 Sonnet, and Qwen2-VL, on standard computer vision tasks like image classification, object detection, semantic segmentation, depth prediction, and surface normal estimation. The authors propose a prompt chaining technique to enable MFMs to solve these tasks, which typically require dense pixel-wise predictions not compatible with the default text output of MFMs. The results show that while MFMs demonstrate respectable generalist abilities, they still lag behind task-specific state-of-the-art vision models in all tasks, with a more pronounced gap in geometric tasks compared to semantic ones. The authors also provide a detailed analysis of the prompt sensitivity and cost associated with their approach.

**Strengths:**

- The topic is meaningful and the presentation is clear and easy-to-follow.

- The authors provide their soulutions to enable MFMs to solve some general CV tasks. These attempts may provide insigths for follow-ups works in unified MFMs.

- This work indicates that although existing MFMs are powerful, they still lag significantly behind CV models in common CV tasks.

- This work also provides an evaluation benchmark for existing MFMs, and we can find GPT4o get the best results for now.

- Reproducibility. The authors provides enough implementation details and algorithm description for the reproducibility.

**Weaknesses:**

- Current designs to support general CV tasks are somewhat inappropriate. For example, current solution for object detection can only handle some easy case, and is likely to give incorrect answers in certain cases (e.g. nearby objects with same category). This issue may lead to some results being unreliable.

- Further discussions and comparisons with other methods are needed. These also some other works provide their solutions to link the MFMs to CV tasks. More experiments to compare the proposed solution and these works are required. For exmaple, I find the authors cite and discuss the DetToolChain, a method to allows MFMs to conduct object detection, in many times (e.g. introduction, related work, even for the semantic segmentation part), but do not provide any comparisons in the object detection section.

- More detailed cost analysis. The proposed methods are generally designed with an iteration manner. Based on this, the cost analysis, with the comparisons of similar prompting methods, is required.

- More ablation experiments. Althought the authors provide extensive experiments, it will be better to provide more ablations for detailed designs in each task.

**Questions:**

Overall, I think the topic of this work is meaningful and may provides insights to the community. However, some designs of prompt chaining are inappropriate, and some results/conclusions may not sufficient. Please see weaknesses for details.

---

### Official Review · Reviewer_Aa1L · 2024-10-28

**Soundness:** 2
**Presentation:** 3
**Contribution:** 2
**Rating:** 5
**Confidence:** 4

**Summary:**

This paper evaluates the performance of several popular multimodal foundation models at standard computer vision tasks. To make MFMs compatible with these tasks, this paper introduces prompt chaining, which splits each task into multiple sub-tasks, and then can be solved in textual prompts. This prompt-chaining framework can be applied to any MFMs with a text interface. By evaluating MFMs on well-established vision tasks and datasets, this paper showcases several observations.

**Strengths:**

(1) The paper is generally well-written and easy to follow.

(2) The experiments are adequate. The framework is tested across multiple tasks and benchmarks, including image classification, semantic segmentation, object detection, depth and surface normal prediction, showcasing its versatility and effectiveness under different settings.

(3) Detailed prompt sensitivity analyses are performed for each task, which is important for MFMs.

(4) Based on the experimental results, several observations are presented.

**Weaknesses:**

(1) **Limited Contribution:** To me, the contribution of this paper is limited. The prompt chaining for each task, which is the critical design in this paper, is naive, commonly used, and fails to achieve satisfactory results. As a research paper, it doesn't propose an effective method for these vision tasks, which severely lags behind the specialist model. As a technique report, the observations presented in this paper are consistent with common sense and don't offer novel insights, resulting in limited value for the community.

(2) **Heavy Cost:** I am concerned about the cost-efficiency of this approach. Though partly effective, the prompt chaining strategies are costly and time-consuming for the multiple API calls for each image, which heavily hinders the practical use, especially for scale inference. This paper didn't compare the time cost between prompt chaining and common prompts, as well as the specialist model.

**Questions:**

(1) **Details of Baseline:** Which prompt is used for naive prompting in Table 8? Is the prompt sensitivity analyzed for these naive prompts? As shown in the paper, there is a varied performance using different prompts. Does the naive prompting always lag behind prompt chaining? Besides, how to set rectangle and curve markers in Figure 12 and Table 12? A common approach for prompting segmentation masks is to use the polygon to capture the boundary (e.g., COCO format). Does it work for MFMs?

(2) **Other Details:** For detection datasets, why choose images containing only a single instance of each present class? It seems that the prompt chaining algorithm for each task depends on several hyper-meters like batchSize, numPairs and weight parameters for final objectives. How to set these values in the experiments, and how do they affect the performance?

---

### Official Review · Reviewer_5rCH · 2024-11-04

**Soundness:** 3
**Presentation:** 3
**Contribution:** 3
**Rating:** 6
**Confidence:** 4

**Summary:**

In this paper, the authors evaluate the visual ability of multimodal foundation models (also known as multimodal large language models) on well-established datasets using the proposed prompt chaining technology. The results show that these models perform better on semantic tasks than geometric tasks. Although the absolute metrics are lower than those of task-specific state-of-the-art models, multimodal models demonstrate potential in processing vision tasks.

**Strengths:**

The proposed prompt chaining technology can evaluate multimodal models on well-established benchmarks by splitting visual tasks into multiple semantic sub-tasks.

Prompt chaining achieves better performance than naive prompting for GPT-4V, suggesting that the inference time scaling law may also apply to visual tasks.

**Weaknesses:**

In Table 2, I notice that Gemini 1.5 Pro with prompt chaining achieves about half the metric of Gemini 1.5 Pro (direct). Can the authors improve the prompt chaining to bridge this significant gap in the future? Alternatively, can the authors further enhance Gemini 1.5 Pro (direct) by refining the prompt chaining technology?

**Questions:**

Can the superpixels be replaced with SAM segmentation/grouping results to achieve better performance on detection or segmentation, or reducing the api cost?

---

### Official Review · Reviewer_ftCV · 2024-11-04

**Soundness:** 3
**Presentation:** 3
**Contribution:** 2
**Rating:** 6
**Confidence:** 2

**Summary:**

This paper evaluates the performance of multimodal foundation models (MFMs) in standard computer vision tasks and it delivers insights about how well the  MFMs can understand pure vision tasks.

**Strengths:**

1. For the first time, this work investigates the vision capabilities of MFMs in standard computer vision tasks by prompt chaining.
2. Extensive experimental results reveal that while MFMs demonstrate general capabilities in handling computer vision tasks, their performance remains below the state-of-the-art.

**Weaknesses:**

1. This paper resembles an engineering study more than a traditional research paper, as its main objective is to assess the performance of MFMs in computer vision tasks.
2. The original design of MFMs, as discussed in the paper, focused on generating improved responses in text-based tasks across multiple modalities. Pure computer vision tasks, however, have their own dedicated foundation models, like SAM for segmentation. Thus, applying existing MFMs to purely computer vision tasks is impractical.

**Questions:**

None

---

### Note · Authors · 2024-11-15

I have read and agree with the venue's withdrawal policy on behalf of myself and my co-authors.